# Neolithic and medieval virus genomes reveal complex evolution of hepatitis B

Ben Krause-Kyora[1,2†]*, Julian Susat[1†], Felix M Key[2], Denise Kühnert[2,3], Esther Bosse[1,4], Alexander Immel[1,2], Christoph Rinne[5], Sabin-Christin Kornell[1], Diego Yepes[4], Sören Franzenburg[1], Henrike O Heyne[6,7,8], Thomas Meier[9,10], Sandra Lösch[11], Harald Meller[12], Susanne Friederich[12], Nicole Nicklisch[12,13], Kurt W Alt[12,13,14,15], Stefan Schreiber[1,16], Andreas Tholey[4], Alexander Herbig[2], Almut Nebel[1], Johannes Krause[2]*

[1]Institute of Clinical Molecular Biology, Kiel University, Kiel, Germany; [2]Max Planck Institute for the Science of Human History, Jena, Germany; [3]Division of Infectious Diseases and Hospital Epidemiology, University Hospital Zurich, Zurich, Switzerland; [4]Systematic Proteomics & Bioanalytics, Institute for Experimental Medicine, Kiel University, Kiel, Germany; [5]Institute of Pre- and Protohistoric Archaeology, Kiel University, Kiel, Germany; [6]Stanley Center for Psychiatric Research, Broad Institute, Cambridge, United States; [7]Analytic and Translational Genetics Unit, Massachusetts General Hospital, Boston, United States; [8]Program in Medical and Population Genetics, Broad Institute of MIT & Harvard, Cambridge, United States; [9]Institute for Pre- and Protohistory and Near Eastern Archaeology, Heidelberg University, Heidelberg, Germany; [10]Heidelberg Center for the Environment, Heidelberg University, Heidelberg, Germany; [11]Department of Physical Anthropology, Institute of Forensic Medicine, University of Bern, Bern, Switzerland; [12]State Office for Heritage Management and Archaeology Saxony-Anhalt, State Museum of Prehistory, Halle, Germany; [13]Danube Private University, Krems, Austria; [14]Department of Biomedical Engineering, University Hospital Basel, University of Basel, Basel, Switzerland; [15]Integrative Prehistory and Archaeological Science, University of Basel, Basel, Switzerland; [16]Clinic for Internal Medicine, University Hospital Schleswig-Holstein, Kiel, Germany

*For correspondence:
b.krause-kyora@ikmb.uni-kiel.de
(BK-K);
krause@shh.mpg.de (JK)

†These authors contributed
equally to this work

Competing interests: The
authors declare that no
competing interests exist.

Reviewing editor: Stephen
Locarnini, Doherty Institute,
Australia

**Abstract** The hepatitis B virus (HBV) is one of the most widespread human pathogens known today, yet its origin and evolutionary history are still unclear and controversial. Here, we report the analysis of three ancient HBV genomes recovered from human skeletons found at three different archaeological sites in Germany. We reconstructed two Neolithic and one medieval HBV genome by *de novo* assembly from shotgun DNA sequencing data. Additionally, we observed HBV-specific peptides using paleo-proteomics. Our results demonstrated that HBV has circulated in the European population for at least 7000 years. The Neolithic HBV genomes show a high genomic similarity to each other. In a phylogenetic network, they do not group with any human-associated HBV genome and are most closely related to those infecting African non-human primates. The ancient viruses appear to represent distinct lineages that have no close relatives today and possibly went extinct. Our results reveal the great potential of ancient DNA from human skeletons in order to study the long-time evolution of blood borne viruses.
DOI: https://doi.org/10.7554/eLife.36666.001

## Introduction

The hepatitis B virus (HBV) is one of the most widespread human pathogens, with worldwide over 250 million people being infected, and an annual death toll of about 1 million globally (*WHO, 2017*). Infection of liver cells with HBV leads to acute hepatitis B, which is self-limiting in about 90–95% of cases. In about 5–10% of infected individuals virus clearance fails and patients develop chronic infection of hepatitis B, which puts them at lifelong elevated risk for liver cirrhosis and liver cancer (hepatocellular carcinoma). HBV is usually transmitted by contact with infectious blood, in highly endemic countries often during birth (*WHO, 2017*).

HBV has a circular, partially double-stranded DNA genome of about 3.2kbp that encodes four overlapping open reading frames (P, pre-S/S, pre-C/C, and X). Based on the genomic sequence diversity, HBVs are currently classified into eight genotypes (A-H) and numerous subgenotypes that show distinct geographic distributions (*Castelhano et al., 2017*). All genotypes are hypothesised to be primarily the result of recombination events (*Littlejohn et al., 2016*; *Simmonds and Midgley, 2005*). To a lesser extent, HBV evolution is also driven by the accumulation of point mutations (*Schaefer, 2007*; *Araujo, 2015*).

Despite being widespread and well-studied, the origin and evolutionary history of HBV are still unclear and controversial (*Littlejohn et al., 2016*; *Souza et al., 2014*). HBVs in non-human primates (NHP), for instance in chimpanzees and gorillas, are phylogenetically closely related to, and yet distinct from, human HBV isolates, supporting the notion of an Africa origin of the virus (*Souza et al., 2014*). Molecular-clock-based analyses dating the origin of HBV have resulted in conflicting estimates with some as recent as about 400 years ago (*Zhou and Holmes, 2007*; *Souza et al., 2014*). These observations have raised doubts about the suitability of molecular dating approaches for reconstructing the evolution of HBV (*Bouckaert et al., 2013* , *Souza et al., 2014*). Moreover, ancient DNA (aDNA) research on HBV-infected mummies from the 16th century AD revealed a very close relationship between the ancient and modern HBV genomes (*Kahila Bar-Gal et al., 2012*; *Patterson Ross et al., 2018*), indicating a surprising lack of temporal genetic changes in the virus during the last 500 years (*Patterson Ross et al., 2018*). Therefore, diachronic aDNA HBV studies are necessary, in which both the changes in the viral genome over time as well as the provenance and age of the archaeological samples are investigated, to better understand the origin and evolutionary history of the virus.

Here, we report the analysis of three complete HBV genomes recovered from human skeletal remains from the prehistoric Neolithic and Medieval Periods in Central Europe. Our results show that HBV already circulated in the European population more than 7000 years ago. Although the ancient forms show a relationship to modern isolates they appear to represent distinct lineages that have no close modern relatives and are possibly extinct today.

## Results and discussion

We detected evidence for presence of ancient HBV in three human tooth samples as part of a metagenomic screening for viral pathogens that was performed on shotgun sequencing data from 53 skeletons using the metagenomic alignment software MALT (*Vågene et al., 2018*). The remains of the individuals were excavated from the Neolithic sites of Karsdorf (Linearbandkeramik [LBK], 5056–4959 cal BC) and Sorsum (Tiefstichkeramik group of the Funnel Beaker culture, 3335–3107 cal BC) as well as from the medieval cemetery of Petersberg/Kleiner Madron (1020–1116 cal AD), all located in Germany (*Figure 1*, *Figure 1—figure supplements 1–3*). After the three aDNA extracts had appeared HBV-positive in the initial virus screening, they were subjected to deep-sequencing without any prior enrichment resulting in 367 to 419 million reads per sample (*Table 1*). A principal component analysis (PCA) of the human DNA recovered from Karsdorf (3-fold genomic coverage) revealed that the sample clusters tightly with other contemporary early Neolithic individuals from the LBK (*Figure 1—figure supplement 4*). The genetic makeup of the early LBK agriculturalists was previously found quite distinct from the preceding western hunter-gatherers of Europe. The genetic shift between both populations was interpreted as a result of early farmers migrating from Western Anatolia into Central Europe introducing agriculture (*Lazaridis et al., 2014*; *Haak et al., 2015*). The almost 2000 years younger Sorsum individual (1.2-fold genomic coverage) clusters in the PCA most closely with individuals from the contemporary Funnel Beaker culture that inhabited

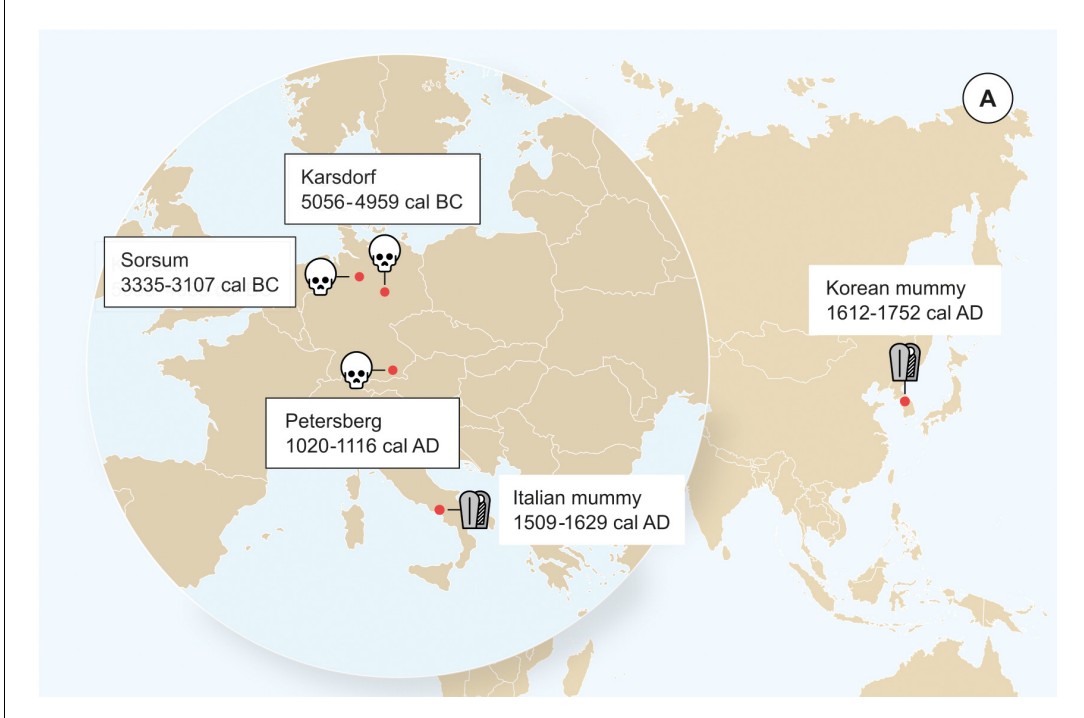

**Figure 1.** Origin of samples. Geographic location of the samples from which ancient HBV genomes were isolated. Radiocarbon dates of the specimens is given in two sigma range. Icons indicate the sample material (tooth or mummy). HBV genomes obtained in this study are indicated by black frame.
DOI: https://doi.org/10.7554/eLife.36666.002

The following figure supplements are available for figure 1:

**Figure supplement 1.** Skull of the investigated Karsdorf individual 537 is from a male with an age at death of around 25–30 years.
DOI: https://doi.org/10.7554/eLife.36666.003

**Figure supplement 2.** Mandible fragment of the Sorsum individual XLVII 11 analyzed in this study is from a male.
DOI: https://doi.org/10.7554/eLife.36666.004

**Figure supplement 3.** Skull of the analyzed Petersberg individual from grave 820 is from a male with an age at death of around 65–70 years.
DOI: https://doi.org/10.7554/eLife.36666.005

**Figure supplement 4.** Principal Component Analysis (PCA) of the human Karsdorf and Sorsum samples together with previously published ancient populations projected on 27 modern day West Eurasian populations (not shown) based on a set of 1.23 million SNPs (*Mathieson et al., 2015*).
DOI: https://doi.org/10.7554/eLife.36666.006

**Figure supplement 5.** Damage plots showing deamination patterns of hg19-specific reads for the HalfUDG-treated libraries of (**a**) Karsdorf, (**b**) Sorsum, (**c**) Petersberg.
DOI: https://doi.org/10.7554/eLife.36666.007

**Figure supplement 6.** Damage plots showing deamination patterns of HBV-specific reads for the HalfUDG-treated libraries of (**a**) Karsdorf, (**b**) Sorsum, (**c**) Petersberg.
DOI: https://doi.org/10.7554/eLife.36666.008

**Figure supplement 7.** MS/MS spectrum of the proteotypic HBV-peptide DLLDTASALYR from the HBV-protein external core antigen (residues 58–68).
DOI: https://doi.org/10.7554/eLife.36666.009

**Figure supplement 8.** Principal Component Analysis (PCA) of the human Karsdorf and Sorsum samples together with previously published ancient populations projected on 27 modern day West Eurasian populations (shown in gray) based on a set of 1.23 million SNPs (*Mathieson et al., 2015*).
DOI: https://doi.org/10.7554/eLife.36666.010

**Figure supplement 9.** Principal Component Analysis (PCA) of the human Petersberg sample projected on 27 modern day West Eurasian populations based on a set of 1.23 million SNPs (*Mathieson et al., 2015*).
DOI: https://doi.org/10.7554/eLife.36666.011

Northern Germany at the end of the fourth millennium BCE (*Figure 1—figure supplement 4*). This population was previously shown to be quite admixed, as a result of a spatial and temporal overlap of early Neolithic farmers and remaining western hunter-gatherers for almost 2000 years (*Bollongino et al., 2013*; *Haak et al., 2015*). The Petersberg individual (2.9-fold genomic coverage),

**Table 1.** Results of the genome reconstruction

| | *Merged reads | Length of HBV consensus sequence | Mean HBV coverage | Gaps in the consensus sequence at nt position | *Mapped reads HBV | *Mapped reads human | Mean human coverage | Human genomes/ HBVgenomes |
|---|---|---|---|---|---|---|---|---|
| Karsdorf | 386,780,892 | 3183 | 104X | 2157–2175; 3107–3128; 3133–3183 | 10,718 | 122,568,310 | 2.96X | 1: 35.1 |
| Sorsum | 367,574,767 | 3182 | 47X | - | 3249 | 9,856,001 | 1.17X | 1: 40.2 |
| Petersberg | 419,413,082 | 3161 | 46X | 880–1000; 1232–1329; 1331–1415; 1420–1581; 1585–1598 | 2125 | 105,476,677 | 2.88X | 1: 16 |

*number.

nt, nucleotide.

DOI: https://doi.org/10.7554/eLife.36666.012

however, showed genetic affinities in the PCA with modern day central European populations. All three ancient human individuals are therefore in agreement with the archeological evidence and radiocarbon dates for their respective time of origin. Together with typical aDNA damage patterns (*Figure 1—figure supplements 5–6*), the human population genetic investigation supports the ancient origin of the obtained datasets.

For successful HBV genome reconstruction, we mapped all metagenomic sequences to 16 HBV reference genomes (eight human genotypes (A-H) and 8 NHPs from Africa and Asia) that are representative of the current HBV strain diversity (*Supplementary file 6*). The mapped reads were used for a *de novo* assembly, resulting in contigs from which one ancient HBV consensus sequence per sample was constructed. The consensus genomes are 3161 (46-fold coverage), 3182 (47-fold coverage), and 3183 (104-fold coverage) nucleotides in length, which falls in the length range of modern HBV genomes and suggests that we successfully reconstructed the entire ancient HBV genomes (*Table 1*, *Figure 2—figure supplements 1–3*). Further, when we conducted liquid chromatography-mass spectrometry (LC-MS) based bottom-up proteomics on tooth material from the three individuals, we identified in the Karsdorf and Petersberg samples a peptide that is part of the very stable HBV core protein, supporting the presence and active replication of HBV in the individuals' blood (*Figure 1—figure supplement 7*).

Phylogenetic network analysis was carried out with a dataset comprised of 493 modern HBV strains representing the full genetic diversity. Strikingly, the Neolithic HBV genomes did not group with any human strain in the phylogeny. Instead, they branched off in two lineages and were most closely related to the African NHP genomes (*Figure 2*, 93% similarity). Although the two Neolithic strains were recovered from humans who had lived about 2000 years apart, they showed a higher genomic similarity to each other than to any other human or NHP genotype. Still, their genomes differed by 6% from each other and may therefore be considered representatives of two separate lineages. They did, however, differ less than 8% from the African NHP strains and should therefore not be called a separate genotype (*Figure 2—figure supplement 4*). The genome from the 1000-year-old Petersberg individual clustered with modern D4 genotypes.

Owing to continuous recombination over time, different gene segments or modules of the ancestral genomes can show up in various subsequent virus generations. Such precursors have been postulated (*Simmonds and Midgley, 2005*) and their existence is supported by the results of our recombination analysis (*Figure 2—figure supplements 5–8*, *Figure 2—source data 1*). Some fragments of the Karsdorf sequences appeared to be very similar to modern human (G, E) and African NHP genotypes, and the Sorsum genome partially showed a high similarity to the human genotypes G, E and B. (*Figure 2—figure supplements 5–8*, *Figure 2—source data 1*). Given the close relationship between the two Neolithic virus genomes, it is also conceivable that the older HBV from Karsdorf could have been a distant source for the younger Sorsum virus (*Figure 2—figure supplements 5–8*, *Figure 2—source data 1*). The closer relationship between the Neolithic and the NHP strains compared to other human strains is noteworthy and may have involved reciprocal cross-species transmission at one or possibly several times in the past (*Simmonds and Midgley, 2005*; *Souza et al., 2014*; *Rasche et al., 2016*).

Taken together, our results demonstrate that HBV already existed in Europeans 7000 years ago and that its genomic structure closely resembled that of modern hepatitis B viruses. Both Neolithic

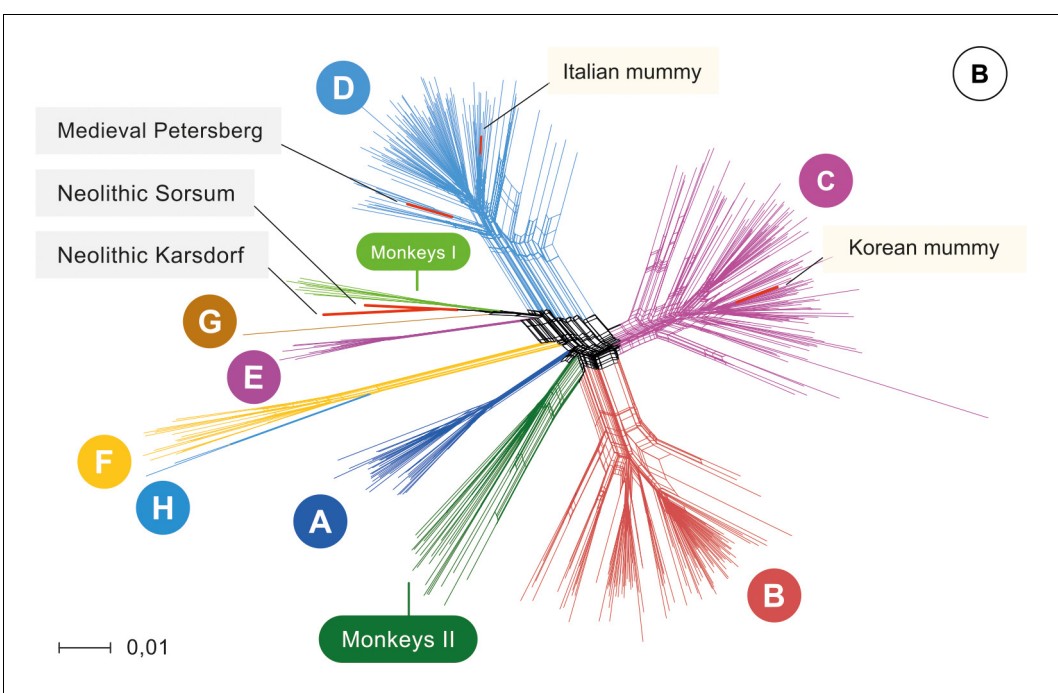

**Figure 2.** Network. Network of 493 modern, two published ancient genomes (light yellow box), and three ancient hepatitis B virus (HBV) obtained in this study (grey box). Colors indicate the eight human HBV genotypes (A–H), two monkey genotypes (Monkeys I, African apes and Monkeys II, Asian monkeys) and ancient genomes (red).
DOI: https://doi.org/10.7554/eLife.36666.013

The following source data and figure supplements are available for figure 2:

**Source data 1.** Results of the recombination analysis using the methods RDP, GENECOV, Chimera, MaxChi, Boot-Scan, SiScan, 3Seq within the RDP v4 software package with all modern full reference genomes (n = 493) and five ancient genomes.
DOI: https://doi.org/10.7554/eLife.36666.023

**Source data 2.** Multiple sequence alignment of the 493 representative and five ancient HBV genomes.
DOI: https://doi.org/10.7554/eLife.36666.024

**Source data 3.** Maximum-likelihood tree based on the multiple sequence alignment of the 493 representative and five ancient HBV genomes with 2000 replicates.
DOI: https://doi.org/10.7554/eLife.36666.025

**Source data 4.** Neighbour-Joining tree based on the multiple sequence alignment of the 493 representative modern and five ancient HBV genomes with 10000 replicates.
DOI: https://doi.org/10.7554/eLife.36666.026

**Figure supplement 1.** Consensus sequence of the Karsdorf HBV genome.
DOI: https://doi.org/10.7554/eLife.36666.014

**Figure supplement 2.** Consensus sequence of the Sorsum HBV genome.
DOI: https://doi.org/10.7554/eLife.36666.015

**Figure supplement 3.** Consensus sequence of the Petersberg HBV genome.
DOI: https://doi.org/10.7554/eLife.36666.016

**Figure supplement 4.** Genetic (hamming) distance of our three ancient HBV genomes compared to all 493 reference genomes.
DOI: https://doi.org/10.7554/eLife.36666.017

**Figure supplement 5.** BootScan analysis of the sequence Karsdorf.
DOI: https://doi.org/10.7554/eLife.36666.018

**Figure supplement 6.** BootScan analysis of the sequence Sorsum.
DOI: https://doi.org/10.7554/eLife.36666.019

**Figure supplement 7.** BootScan analysis of the sequence Petersberg.
DOI: https://doi.org/10.7554/eLife.36666.020

**Figure supplement 8.** SimPlot analysis of (**a**) Karsdorf, (**b**) Sorsum and (**c**) Petersberg.
DOI: https://doi.org/10.7554/eLife.36666.021

*Figure 2 continued on next page*

*Figure 2 continued*

**Figure supplement 9.** Plot of phylogenetic root-to-tip distance relative to sampling time (TempEst).
DOI: https://doi.org/10.7554/eLife.36666.022

viruses fall between the present-day modern human and the known NHP diversity. Therefore, it can be hypothesized that although the two Neolithic HBV strains are no longer observed today and thus may reflect two distinct clades that went extinct, they could still be closely related to the remote ancestors of the present-day genotypes, which is supported by signs of ancient recombination events. More ancient precursors, intermediates and modern strains of both humans and NHPs need to be sequenced to disentangle the complex evolution of HBV. As this evolution is characterized by recombination and point mutations and may further be complicated by human-ape host barrier crossing (*Simmonds and Midgley, 2005*; *Souza et al., 2014*; *Rasche et al., 2016*), genetic dating is not expected to yield meaningful results. This is additionally supported by a TempEst analysis (*Rambaut et al., 2016*) that shows very little temporal signal (*Figure 2—figure supplement 9*). It should, however, be noted that the oldest genome (Karsdorf) was found in an individual that belonged to a population of early farmers that had migrated in the previous few hundred years from the Near East into central Europe. One might speculate that the close proximity to recently domesticated animals, changes in subsistence strategy as well as the adopted sedentary lifestyle might have contributed to the spread of HBV within Neolithic human populations.

Based on our analysis, HBV DNA can reliably be detected in tooth samples that are up to 7000 years old. Ancient HBV has so far only been identified in soft tissue from two 16[th]-century mummies (*Kahila Bar-Gal et al., 2012*; *Patterson Ross et al., 2018*). The aDNA analysis of HBV from prehistoric skeletons, which facilitates evolutionary studies on a far-reaching temporal scale, has not been described up to now. One explanation for the difficulty of a molecular HBV diagnosis in bones is that the virus infection does not leave lesions on skeletal remains that would allow researchers to select affected individuals *a priori*, as it is the case for instance for leprosy (*Schuenemann et al., 2013*). The diagnosis of an HBV infection in skeletal populations is purely a chance finding and is thus more probable in a large-scale screening.

Overall, HBV biomolecules seem to be well preserved in teeth: Avoiding biases from DNA capture and reference-based mapping we could reconstruct three HBV genomes by *de novo* assembly from shotgun data and even observed HBV-specific peptides. The ratio of HBV genomes to the human genome in our samples was rather high and similar in all three samples (Karsdorf 35:1, Sorsum 40.2:1 and Petersberg 16:1). As there is no evidence that HBV DNA is more resistant to postmortem degradation than human DNA, the high rate of HBV compared to human DNA may reflect the disease state in the infected individuals at the time of death. High copy numbers of viral DNA in the blood of infected individuals are associated with acute HBV infection, or reactivation of chronic HBV. Thus, it seems likely that the death of the ancient individuals is related to the HBV infection, but might not be the direct cause of death as fulminant liver failure is rather rare in modern day patients. The HBV infection might have instead contributed to other forms of lethal liver failure such as cirrhosis or liver cancer.

In view of the unexpected complexity of our findings, we envisage future diachronic HBV studies that go beyond the temporal and geographic scope of our current work.

## Materials and methods

### Human remains

The LBK settlement of Karsdorf, Saxony-Anhalt, Germany, is located in the valley of the river Unstrut. Between 1996 and 2010 systematic excavations were conducted at Karsdorf that led to the discovery of settlements and graves from the Neolithic to the Iron Age (*Behnke, 2007*; *2011*; *2012*). The LBK is represented by 24 longhouses in north-west to south-east orientation that were associated with settlement burials (*Veit, 1996*). The investigated individual 537 is a male with an age at death of around 25–30 years (*Figure 1—figure supplement 1*), dated to 5056–4959 cal BC (KIA 40357–6116 ± 32 BP) (*Brandt et al., 2014*; *Nicklisch, 2017*).

The gallery grave of Sorsum, Lower-Saxony, Germany, is typologically dated to the Tiefstichkeramik (group of the Funnelbeaker culture). Sorsum is exceptional as it was built into the bedrock. During the excavations (1956–1960) of the grave chamber around 105 individuals were recovered (*Claus, 1983*; *Czarnetzki, 1966*). Individual XLVII 11 analyzed in this study is a male (*Figure 1—figure supplement 2*) and dates to 3335–3107 cal BC (MAMS 33641–4501 ± 19 BP).

The medieval cemetery of Petersberg/Kleiner Madron, Bavaria, Germany, lies on a hill top at 850 meters asl and 400 meters above the floor of the Inn Valley. On the eastern part of the cemetery members of a priory were buried that was most likely established in the late 10th century. Written sources document its existence from 1132 onwards (*Meier, 1998*). During systematic excavations (1997–2004) in the southeastern part of the churchyard the remains of individuals buried in 99 graves were uncovered. The examined individual in grave 820 is a male with an age at death of around 65–70 years (*Lösch, 2009* - *Figure 1—figure supplement 3*) dating to 1020–1116 cal AD (MAMS 33642–982 ± 17 BP).

## DNA extraction and sequencing

The DNA extractions and pre-PCR steps were carried out in clean room facilities dedicated to aDNA research. Teeth were used for the analyses. The samples from Petersberg and Sorsum were processed in the Ancient DNA Laboratory at Kiel University and the sample from Karsdorf in the Ancient DNA Laboratory of the Max Planck Institute for the Science of Human History (MPI SHH) in Jena. All procedures followed the guidelines on contamination control in aDNA studies (*Warinner et al., 2017*; *Key et al., 2017*). The teeth were cleaned in pure bleach solution to remove potential contaminations prior to powdering. Fifty milligrams of powder were used for extraction following a silica-based protocol (*Dabney et al., 2013*). Negative controls were included in all steps.

From each sample, double-stranded DNA sequencing libraries (UDGhalf) were prepared according to an established protocol for multiplex high-throughput sequencing (*Meyer and Kircher, 2010*). Sample-specific indices were added to both library adapters via amplification with two index primers. Extraction and library blanks were treated in the same manner. For the initial screening, the library of the individual from Karsdorf was sequenced on 1/50 of a lane on the HiSeq 3000 (2 × 75 bp) at the MPI SHH in Jena and the libraries from Petersberg and Sorsum were sequenced on the Illumina HiSeq 4000 (2 × 75 bp) platform at the Institute of Clinical Molecular Biology, Kiel University, using the HiSeq v4 chemistry and the manufacturer's protocol for multiplex sequencing. Deep-sequencing for each of the three samples was carried out on two lanes on the Illumina HiSeq 4000 platform at the Institute of Clinical Molecular Biology, Kiel University.

## Clip and merge

The datasets produced for all ancient samples contained paired-end reads with varying numbers of overlapping nucleotides as well as artificial adapter sequences. We used ClipAndMerge version 1.7.3, a module of the EAGER pipeline (*Peltzer et al., 2016*), to clip adapter sequences, merge corresponding paired-end reads in overlapping regions and to trim the resulting reads. We used the default options with the following command:

```
java -jar ClipAndMerge.jar -in1 $FASTQ1 -in2 $FASTQ2 \
-f AGATCGGAAGAGCACACGTCTGAACTCCAGTCAC \
-r AGATCGGAAGAGCGTCGTGTAGGGAAAGAGTGTA \
-l 25 -qt -q 20 -o $output_file
where $FASTQ1 and $FASTQ2 are the two gzipped FASTQ input files
```

## Adapter clipping

ClipAndMerge uses an overlap alignment of the respective forward or reverse adapter with the 3' end of each read in order to remove sequencing adapter sequences. Regions at the 3' end of each read that were contained in the alignment were clipped. Reads that were shorter than 25 nucleotides after adapter clipping or contained only adapter sequences (adapter dimers) were removed. All remaining reads were then used in the merging step.

## Merging of overlapping paired reads

Merging was performed for all remaining paired reads with a minimum overlap of 10 nucleotides and at most 5% mismatches in the overlap region. The algorithm selected the maximal overlap fulfilling these criteria. The consensus sequence was generated using the nucleotides in the overlap regions from the read with the higher PHRED quality score, maximizing the quality of the resulting read.

## Quality trimming

In a final step, ClipAndMerge performed quality trimming of the reads and all nucleotides with PHRED scores smaller than 20 were trimmed from the 3' end of each read. Finally, all reads with fewer than 25 nucleotides after quality trimming were removed. The resulting high-quality reads were used for the alignment.

## Virus screening

Screening of the datasets was carried out with the software MALT using the ncbi-viral database as a reference. A sequence identity threshold of 85% was set and the alignment mode was changed to SemiGlobal. The analysis was carried out using the following command:

malt-run –mode BlastN -e 0.001 -id 85 –alignmentType SemiGlobal –index $index –inFile $FASTQCM –output $OUT

where $index is the index file, $FASTQCM is the clipped and merged file and $OUT is the output the file.

The resulting alignments were visually inspected using MEGAN 6. Reads mapping to the hepatitis B reference in the database (NC_003977.2) were extracted and verified using a discontiguous megablast against the virus taxa (taxid: 10239) with default parameters.

## HBV alignment

For identification of the genotypes, samples were aligned against one reference for each of the eight hepatitis B genotypes available in the NCBI hepatitis B genotyping project (https://www.ncbi.nlm.nih.gov/projects/genotyping/view.cgi?db=2) (*Supplementary file 1*). Additionally, eight NHP strains were used. All references were combined in one FASTA file and a competitive mapping was performed using BWA. The mapping was carried out using the following command:

bwa aln -n 0.01 l 300 $INDEX $FASTQCM $OUT

where $INDEX is the reference, $FASTQCM is the input file and $OUT is the output file.

Minimum mapping quality was set to 0.

## Duplicate removal

We used DeDup version 0.11.3, part of the EAGER pipeline,*Peltzer et al., 2016* to identify and remove all duplicate reads in the sample specific BAM files (*Supplementary file 2*) with the default options and the following command:

java -jar DeDup.jar -i $IN -o $OUT

where $IN is the input BAM file and $OUT is the output BAM file.

## Extracting mapped reads

After duplicate removal the resulting BAM files were converted to SAM files using SAMtools version 0.1.19-96b5f2294a with default parameters and the following command:

samtools view -h -o $OUT $IN

where $OUT is the SAM output file and $IN is the BAM input file. Reads from the SAM file where converted to FASTQ using the following awk script:

awk '/[FMR]/{print '@"$1'\n'$10'\n+\n'$11}' $IN > $OUT where $IN is a SAM file and $OUT is the resulting FASTQ file containing all the mapped reads.

## De novo assembly

The de novo assembly was performed using the SPAdes genome assembler version v3.9.0 (*Bankevich et al., 2012*) with the following settings:

spades.py -t 20 m 500 k

11,13,15,17,19,21,23,25,27,29,31,33,35,37,39,41,43,45,47,49,51,53,55,57,59,61,63,65,67,69,71,
73,75,77,79,81,83,85,87,89,91,93,95,97,99,101,103,105,107,109,111,113,115,117,119,121,123,125,
127 s $IN -o $OUT

where $IN is a FASTQ file containing the mapped reads and $OUT is the output folder for
SPAdes.

Resulting contigs for each K-value were checked and the one which spawned the longest contig
was selected for further processing (*Supplementary file 3*).

## Mapping of contigs

Contigs were mapped against the multi FASTA file containing all 16 references. The following command was used:

bwa mem $INDEX $IN $OUT

where $INDEX is the reference, $IN the file containing the contig/contigs and $OUT is the resulting BAM file.

## Consensus generation

For genomic reconstruction of the ancient HBV strains, the results of the alignments were inspected visually with IGV version 2.3.92 (*Thorvaldsdóttir et al., 2013*). Information about contig order and direction were used for the construction of a consensus sequence. Bases that were soft clipped in the alignment were cut off using SeqKit software version 0.7.0 and realigned to the 16 references as described above. This was done because of the circular genome structure of HBV. Big contigs needed to be split to preserve genomic order with respect to the reference sequences (*Supplementary file 4*).

## Remapping raw reads against the consensus sequence

Raw reads of each sample were mapped to their corresponding consensus sequence using the software CircularMapper version 1.93.4 and the following command line:

java -jar CircularGenerator.jar -e $E -i $IN -s '$N'

where $E is the length of elongation, $IN is the input file and $N is the name of the target sequence.

bwa aln -t 8 $IN $R -n 0.01-l 300-f $OUT

where $IN is the elongated consensus sequence, $R is the file containing the clipped and merged reads and $OUT is the output file.

bwa samse $RE $IN $R -f $OUT

where $RE is the elongated reference, $IN is the bwa aln output, $ R is the file containing the clipped and merged reads and $OUT is the output file.

java -jar realign-1.93.4.jar -e $E -i $IN -r $OR

where $E is the length of elongation, $IN is the output of bwa samse and $OR is the unmodified consensus.

## Phylogenetic analysis

Hepatitis B reference strains for apes were collected using edirect with the following command:

esearch -db pubmed -query 'hepatitis B AND Orangutan OR hepatitis B AND Gibbon OR hepatitis B AND Gorilla OR hepatitis B AND Chimpanzee OR hepatitis B AND Orang-utan' | elink -target nuccore | efetch -format fasta > $OUT

where $OUT is the output file in fasta format containing all sequences from the papers containing the search keys.

To control the received sequences a multiple sequence alignment using the linsi algorithm contained in MAFFT version 7.310 was carried out. The following command was used:

linsi $IN > $OUT

where $IN is the input file containing the retrieved sequences and $OUT is the multiple sequence alignment.

The alignment was visually inspected in AliView (v. 1.18.1) and sequences that differed from the majority were removed. This step was necessary due to the unrestricted esearch command which, by

chance, could also return non-primate sequences. After filtering the set contained 74 ape infecting HBV strains.

Using the 74 ape strains and 5497 non-recombinant genomes available at hpvdb (https://hbvdb. ibcp.fr/HBVdb/HBVdbDataset?seqtype=0) clustering was carried out with UClust v 1.1.579 (*Edgar, 2010*). The clustering with an identity threshold of 97% yielded 493 representative HBV genomes. Combining them with the five ancient strains a multiple sequence alignment was carried out using Geneious version 10.1.2 (*Kearse et al., 2012*) with a 65% similarity cost matrix, a gap open penalty of 12 and a gap extension penalty of 3. The multiple sequence alignment was stripped of any sites (columns) that had gaps in more than 95% of sequences. The complete alignment including all modern and ancient genomes as multi-fasta is available in *Figure 2—source data 2*. The alignment was used to construct a network with the software SplitsTree v4 (*Huson and Bryant, 2006*), creating a NeighborNet (*Bryant and Moulton, 2004*) with uncorrected P distances.

The same multiple sequence alignment was used for the generation of Maximum-Likelihood (ML) and Neighbour-Joining (NJ) Trees. MEGA7 version 7170509-x86_64 with the following command line was used:

Megacc -a $MAO -d $IN -o $OUT

where $MAO is the megacc configuration file, $IN is the multiple alignment and $OUT is the output directory. For both trees 1408 informative sites and Jukes-Cantor substitution model were used. Bootstrap replicates are 2000 for ML and 10000 for NJ. The trees are provided in *Figure 2—source data 3* and *4*.

## Molecular clock analysis

The evolution of hepatitis B virus over time is unclear with regard to its evolutionary rate and the role of recombination. Previous studies have attempted to detect a molecular clock-like signature without success. We investigate if the ancient genomes presented here allow a molecular clock analysis using TempEst v1.5.1 (*Rambaut et al., 2016*). The data set shows little positive correlation between genetic divergence and sampling time (correlation coefficient 0.075) and there is very little temporal signal (TempEst $R2 = 0.006$, see *Figure 2—figure supplement 9*). Therefore, we refrain from further dating analysis.

## Recombination analysis

We performed recombination analysis using all modern full reference genomes (n = 493) and five ancient genomes used for the network analysis (see above in Phylogenetic analysis). The methods RDP, GENECOV, Chimera, MaxChi, BootScan, SiScan, 3Seq within RDP v4 (*Martin et al., 2015*) with a window size of 100 nt and the parameter set to circular genome with and without outgroup reference (results are provided in *Figure 2—source data 1*) and SimPlot v 3.5.1 (*Lole et al., 1999*, *Figure 2—figure supplements 5–8*) were applied to the data set.

## LC-MS/MS analysis and database searches

Proteins were extracted from powdered tooth samples (50 mg) using a modified filter-aided sample preperation (FASP) protocol as previously described (*Warinner et al., 2014*; *Cappellini et al., 2014*). In-filter trypsin digested samples were analyzed on a Dionex Ultimate 3000 nano-HPLC coupled to a Q Exactive mass spectrometer (Thermo Scientific, Bremen). The samples were washed on a trap column (Acclaim Pepmap 100 C18, 10 mm ×300 µm, 3 µm, 100 Å, Dionex) for 5 min with 3% acetonitrile (ACN)/0.1% TFA at a flow rate of 30 µL/min prior to peptide separation on an Acclaim PepMap 100 C18 analytical column (15 cm ×75 µm, 3 µm, 100 Å, Dionex). A flow rate of 300 nL/min using eluent A (0.05% formic acid (FA)) and eluent B (80% ACN/0.04% FA) was used for gradient separation as follows: linear gradient 5 ± 50% B in 60 min, 50 ± 95% B in 5 min, 95% B for 10 min, 95 ± 5% B in 1 min, and equilibration at 5% B for 12 min. Spray voltage applied on a metal-coated Pico-Tip emitter (30 µm tip size, New Objective, Woburn, Massachusetts, US) was 1.25 kV, with a source temperature of 250℃. Full scan MS spectra were acquired from 5 to 145 min between 300 and 2,000 m/z at a resolution of 60,000 at m/z 400 (automatic gain control [AGC] target of 1E6; maximum ion injection time [IIT] of 500 ms). The five most intense precursors with charge states 2 + used were selected with an isolation window of 1.6 m/z and fragmented by HCD with normalized collision

energies of 25. The precursor mass tolerance was set to 10 ppm, and dynamic exclusion (30 s) was enabled.

Acquired spectra were analyzed by database searches using Proteome Discoverer (PD) 2.2.0.388 with the search engines SequestHT (Thermo Scientific). Searches were performed against a combined database built by the combination of the full Swiss protein database (468,716 entries, downloaded from Uniprot, December, 21$^{st}$, 2017), hepatitis B data base (seven entries, downloaded from Uniprot, December, 7$^{th}$, 2017) and common laboratory contaminants (115 entries, downloaded from Uniprot, August, 15$^{th}$, 2014). The following settings were used for the search: semi-tryptic specificity; two missed cleavage sites; mass tolerances of 10 ppm for precursors and for fragment masses 0.02 Da (HCD) and 0.5 Da (CID); static modifications: carbamidomethylation on Cys; dynamic modifications: oxidation of Met, Lys and Pro. An additional search was performed using 12 FASTA files from in silico translated DNA sequences. The DNA sequences were obtained from previous DNA sequencing of the samples.

A nearly complete y-ion series and two b-ion fragments allow for an assignment of the full peptide sequence. The peptide was identified in the biological sample from Petersberg with four peptide spectral matches, showing that the detection of this peptide is not a random event. Moreover, the same peptide could also be identified in the second biological sample from Karsdorf (not shown); blank runs between the LC-MS/MS runs of the two samples rule out potential artifacts due to sample carryover.

Note that the MS/MS method applied here does not allow us to distinguish leucine (L) or isoleucine (I) residues. Manual permutation of the leucine residues in the above stated sequence followed by a BLAST search (default search parameters) led to the identification of the HBV-protein external core antigen in all cases with the exception of the combinations D*II*DTASALYR and DLLDTASA*I*YR; these two variants were reported by BLAST search as the proteins hypothetical protein CR988_04570 [*Treponema sp.*] and anti-GFP antibody [synthetic construct] with the HBV-protein external core antigen listed at rank 3. However, these proteins were not found in the genomic data. Hence, despite the uncertainty of the I/L assignment, the MS/MS data support the genomic finding of an HBV infection.

## Human population genetic analyses

Mapping of the adapter-clipped and merged FASTQ files to the human reference genome hg19 was done using BWA (*Li and Durbin, 2010*) with the following command line:

bwa aln -n 0.01 l 300 $INDEX $FASTQCM $OUT

where $INDEX is the reference, $FASTQCM is the input file and $OUT is the output file. The duplicate removal after mapping was executed as described above.

The mapped sequencing data were transformed into the *Eigenstrat* format (*Price et al., 2006*) and merged with a dataset of 1,233,013 SNPs (*Haak et al., 2015*; *Mathieson et al., 2015*). Using the software Smartpca (*Patterson et al., 2006*) the three samples and previously published ancient populations were projected onto a base map of genetic variation calculated from 32 West Eurasian populations (*Figure 1—figure supplements 4*, *8* and *9*).

## Sex determination

Sex was assessed based on the ratio of sequences aligning to the X and Y chromosomes compared to the autosomes (*Skoglund et al., 2013*).

## Acknowledgements

We are grateful to the following people and institutions for providing samples, support, and advice: Bodo Krause-Kyora, Hildegard Nelson (Referat A1 Archäologische Dokumentation, Niedersächsisches Landesamt für Denkmalpflege), Ulrike Weller (Sammlungsverwaltung Archäologie Landesmuseum Hannover) and Britta Steer for technical assistance with proteomics sample preparation. This work was supported by the Collaborative Research Centre 1266 *Scales of Transformation*, the Excellence Cluster 306 *Inflammation at Interfaces*, the Medical Faculty of Kiel University, the Max Planck Society and the European Research Council (ERC) starting grant APGREID (to JK). Excavations and analysis of the archaeological site of Karsdorf were supported by the German Research Foundation (DFG) Grant of Kurt W Alt (Al 287-7-1) and Harald Meller (Me 3245/1–1). Analysis of the

archaeological site of Petersberg/Kleiner Madron was supported by a grant of the VolkswagenStiftung to Thomas Meier and by a PhD-Fellowship of the Ludwig-Maximilians-University, Munich to Sandra Lösch.

## Additional information

### Funding

| Funder | Grant reference number | Author |
| --- | --- | --- |
| Collaborative Research Center | 1266 | Ben Krause-Kyora<br>Almut Nebel |
| Swiss National Science Foundation | PMPDP3_171320/1 | Denise Kühnert |
| Deutsche Forschungsgemeinschaft | HE7987/1-1 | Henrike O Heyne |
| Deutsche Forschungsgemeinschaft | Me 3245/1-1 | Harald Meller |
| Deutsche Forschungsgemeinschaft | AI 287-7-1 | Kurt W Alt |
| European Research Council | APGREID | Johannes Krause |

The funders had no role in study design, data collection and interpretation, or the decision to submit the work for publication.

### Author contributions

Ben Krause-Kyora, Conceptualization, Resources, Data curation, Formal analysis, Supervision, Funding acquisition, Validation, Investigation, Methodology, Writing—original draft, Project administration, Writing—review and editing; Julian Susat, Felix M Key, Alexander Herbig, Data curation, Formal analysis, Investigation, Writing—original draft, Writing—review and editing; Denise Kühnert, Esther Bosse, Formal analysis, Investigation; Alexander Immel, Formal analysis, Investigation, Writing—original draft; Christoph Rinne, Harald Meller, Susanne Friederich, Stefan Schreiber, Resources; Sabin-Christin Kornell, Investigation; Diego Yepes, Sören Franzenburg, Formal analysis; Henrike O Heyne, Formal analysis, Writing—original draft; Thomas Meier, Sandra Lösch, Resources, Investigation; Nicole Nicklisch, Resources, Formal analysis, Investigation; Kurt W Alt, Resources, Formal analysis; Andreas Tholey, Resources, Supervision, Methodology, Writing—original draft; Almut Nebel, Resources, Formal analysis, Funding acquisition, Writing—original draft, Writing—review and editing; Johannes Krause, Conceptualization, Resources, Data curation, Supervision, Funding acquisition, Methodology, Writing—original draft, Project administration, Writing—review and editing

### Author ORCIDs

Ben Krause-Kyora http://orcid.org/0000-0001-9435-2872
Felix M Key http://orcid.org/0000-0003-2812-6636
Johannes Krause http://orcid.org/0000-0001-9144-3920

### Ethics

Human subjects: The human remains are prehistoric European specimens, so consent was not required. No decedent groups claim responsibility or ancestry to those people.

### Decision letter and Author response

Decision letter https://doi.org/10.7554/eLife.36666.037
Author response https://doi.org/10.7554/eLife.36666.038

## Additional files

### Supplementary files
• Supplementary file 1. Accession numbers for the reference genomes used in the first alignment step to catch HBV diversity in the sample. Since monkey HBV strains are not classified into genotypes the column is left blank.
DOI: https://doi.org/10.7554/eLife.36666.027

• Supplementary file 2. Number of reads mapping against the references shown in *Supplementary file 1* before and after duplicate removal.
DOI: https://doi.org/10.7554/eLife.36666.028

• Supplementary file 3. Number of contigs and combined contig length of the de novo assembly for chosen K-values.
DOI: https://doi.org/10.7554/eLife.36666.029

• Supplementary file 4. Final consensus length after retrieving gap information from the multiple sequence alignment with Geneious.
DOI: https://doi.org/10.7554/eLife.36666.030

• Supplementary file 5. Number of reads mapping against hg19 before and after duplicate removal and percentage of the genome where coverage is at least one.
DOI: https://doi.org/10.7554/eLife.36666.031

• Supplementary file 6. Basic statistics for the mapping against the references shown in table S1. Shown are mean coverage, mean coverage for the covered region, genome length, number of missing bases and covered bases
DOI: https://doi.org/10.7554/eLife.36666.032

• Transparent reporting form
DOI: https://doi.org/10.7554/eLife.36666.033

### Data availability
Raw sequence read files have been deposited at the European Nucleotide Archive under accession no. PRJEB24921

The following dataset was generated:

| Author(s) | Year | Dataset title | Dataset URL | Database, license, and accessibility information |
| --- | --- | --- | --- | --- |
| Krause J | 2018 | High-throughput sequence data for Neolithic and Medieval virus genomes reveal complex evolution of Hepatitis B | https://www.ebi.ac.uk/ena/data/view/PRJEB24921 | Publicly available at the European Nucleotide Archive (accession no. PRJEB24921) |

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
