## [Decision Letter]

Thank you for submitting your article "Neolithic and Medieval virus genomes reveal complex evolution of Hepatitis B" for consideration by *eLife*. Your article has been favorably evaluated by Wenhui Li (Senior Editor) and three reviewers, one of whom is a member of our Board of Reviewing Editors. The following individual involved in review of your submission has agreed to reveal their identity: Stefan Seitz (Reviewer #3).

The reviewers have discussed the reviews with one another and the Reviewing Editor has drafted this decision to help you prepare a revised submission.

Summary:

Research into the origin and evolution of HBV has been on the increase in recent years, and has probably been encouraged by the recovery of HBV from 400-year-old Korean and Italian mummies (Kahila Bar-Gal et al., 2012 and Patterson Ross et al., 2018). Nonetheless, the origin of human and non-human primate HBV remains in debate. Thus, the recovery of HBV from Neolithic and Medieval archaeological remains reported in this manuscript by Krause-Kyora et al. will make a significant contribution to this field. This manuscript describes the reconstruction of three ancient HBV genomes from the teeth of two Neolithic (5 and 7 thousand years old, kya) and one Medieval (1 kya) individuals from Germany. The two Neolithic HBV genomes clustered phylogenetically with, but distinctly from, African non-human primate HBV sequences, and the Medieval HBV clustered with contemporary HBV/D4 genome sequences. Importantly, the authenticity of these HBV was also confirmed by the detection of a peptide of HBV core protein from two of the samples.

Overall, this manuscript is very interesting and timely. The HBV's recovered are not only the oldest to date, but the two more ancient viruses actually clustered with African non-human primate HBV is a remarkable finding. Especially since the tooth samples were of individuals who have existed in the region around 2,000 years apart, and the two ancient HBV genomes were only 6% different genetically. We have the following suggestions to improve the manuscript.

Essential revisions:

1) The three ancient HBV genomes recovered are novel and need to be properly characterised in accordance to HBV taxonomy, and must be discussed and presented in the supplement. For example, do the genomes of these ancient HBV have the four overlapping open reading frames and key regulatory elements that are present in the modern human and non-human primate HBV's (e.g. epsilon motif, YMDD motif in polymerase ORF, 'a'-determinant in the surface antigen, etc.)? In addition, non-human primate HBV and HBV genotype D share a 33nt deletion within the preS1 domain (Takahashi et al. Virology 2000). Do the authors see a similar pattern with the ancient HBV sequences? The genetic distance between two older HBV's is less than 8% and can be considered the same 'genotype'. What is the mean genetic distance between these HBV's and the African non-human primate HBV's, and do the authors believe that these two ancient HBV's are sufficient to be designated as a novel genotype? Were the authors able to predict the HBeAg status (i.e. based on mutations in the HBV basal core promotor and/or precore gene) of the infected individuals from the sequence? This would give some important hints concerning the status of the infection.

2) There is little discussion on the phylogenetic network presented except stating the two older HBV's clustered with the African non-human primate HBV. Was there any indication of evolutionary events such as recombination? Although phylogenetic networks can describe a wider range of evolutionary events than phylogenetic trees, their ability and accuracy for detecting recombination events have not be proven formally. Thus, NJ, ML or Bayesian tree analyses with well-defined reference sequences are needed. We also suggest generating phylogenetic trees with different regions of the HBV genome, in accordance to the SimPlot results. Trees should show bootstrap values or posterior probabilities to support the robustness of the phylogeny.

3) The manuscript would also benefit if more detail on sampling were provided. What was the total number of archaeological sites studied, and for each site, the proportion of tooth samples that was human DNA positive and the proportion of these human DNA positive samples that were also HBV DNA positive?

4) The authors should put the ancient HBV's discovered into a human history context for the readers, especially since principle component analysis was performed to compare the human DNA genomes recovered in this study with other ancient and modern human genomes (Figure 1—figure supplement 4, the source of these datasets need to be acknowledged). What was the historical background of the people in the region at around the time frames studied? This should be discussed in light of the current state of knowledge about human population genetics and paleogenomics of European people, namely the work of Lazaridis et al., 2014. The authors should also compare the HBV "genotypes" that were present at around 7,000 years ago to those that are present in Germany now.

5) It has been shown in numerous studies that avihepadnaviruses are able to integrate into their host genomes, via the germ line, to form endogenous copies. Integration outside of hepatocytes has not been demonstrated for HBV. Nonetheless, in the case of this study we would appreciate a rationale for why the ancient HBV sequences represent exogenous virion DNA rather than endogenous elements.

6) The HBV mutation rate has been a matter of extensive debate over recent years and studies into ancient HBV sequences dating back a few centuries have failed to detect any clock-like signature. These three much older HBV sequences are an exceptional opportunity to assess the evolution of HBV more accurately over time. The manuscript would benefit greatly from some molecular clock analysis.

7) The authors should discuss what significance these ancient HBV sequences have on the origin of orthohepadnaviruses in humans? What changes result from the inclusion of these three ancient sequences in the phylogenetic tree?

8) Considering double infection (super/coinfection) with HBV of two different genotypes is a common phenomenon, with prevalence rates ranging from 4-18% (Ding et al., 2003; Kato et al., 2003; Osiowy and Giles 2003; Chen et al., 2004; Olinger et al., 2006), the authors should comment on their certainty that the consensus sequence recovered from each of the ancient samples do represent one HBV strain and not an artificial chimeric HBV sequence, the result of contigs generated from short reads of next-generation sequencing of two different HBV strains circulating within the same infected individual? The authors have discussed the evidence for extensive recombination, but did not discuss whether this may stem from double infections.

---

## [Author Response]

Essential revisions:1) The three ancient HBV genomes recovered are novel and need to be properly characterised in accordance to HBV taxonomy, and must be discussed and presented in the supplement. For example, do the genomes of these ancient HBV have the four overlapping open reading frames and key regulatory elements that are present in the modern human and non-human primate HBV's (e.g. epsilon motif, YMDD motif in polymerase ORF, 'a'-determinant in the surface antigen, etc.)? In addition, non-human primate HBV and HBV genotype D share a 33nt deletion within the preS1 domain (Takahashi et al. Virology 2000). Do the authors see a similar pattern with the ancient HBV sequences? The genetic distance between two older HBV's is less than 8% and can be considered the same 'genotype'. What is the mean genetic distance between these HBV's and the African non-human primate HBV's, and do the authors believe that these two ancient HBV's are sufficient to be designated as a novel genotype? Were the authors able to predict the HBeAg status (i.e. based on mutations in the HBV basal core promotor and/or precore gene) of the infected individuals from the sequence? This would give some important hints concerning the status of the infection.

Thank you for the suggestion, we characterized the new genomes according to HBV taxonomy and presented them in the supplements (Figure 2—figure supplements 1, 2 and 3). In addition, we could identify in the Karsdorf and Sorsum genomes the epsilon motif, YMDD motif in polymerase ORF, 'a'-determinant in the surface antigen. Further the 33nt deletion is present in both genomes. We calculated the hamming distance between our ancient genomes and all reference genomes (genotype A-H and the non-human primate HBV's). Both Neolithic HBV genomes, Sorsum and Karsdorf, have a genetic distance to the African non-human primate HBV of under 8% (Figure 2—figure supplement 4). Thus, it is not sufficient to support a new genotype. We added a sentence to the third paragraph of the Results and Discussion. The ancient HBV genome from Petersberg has, as expected, only low divergence to genotype D HBV genomes, supporting its phylogenetic placement. The coverage plots indicate a higher copy number of the HBeAg coding sequence. This could be due to differential preservation of parts of the DNA, single stranded vs. double stranded DNA or secondary structures of the DNA. Due to the limitation of ancient DNA research, e.g. preservation conditions of the DNA or degradation processes, the quantification is challenging and did not allow any proof of the status of infection. We do discuss however that the high copy number of HBV versus human DNA might indicate any acute infections (Results and Discussion, seventh paragraph).

2) There is little discussion on the phylogenetic network presented except stating the two older HBV's clustered with the African non-human primate HBV. Was there any indication of evolutionary events such as recombination? Although phylogenetic networks can describe a wider range of evolutionary events than phylogenetic trees, their ability and accuracy for detecting recombination events have not be proven formally. Thus, NJ, ML or Bayesian tree analyses with well-defined reference sequences are needed. We also suggest generating phylogenetic trees with different regions of the HBV genome, in accordance to the SimPlot results. Trees should show bootstrap values or posterior probabilities to support the robustness of the phylogeny.

We attached the NJ and ML tree analysis as Figure 2—source data 3 and 4. The NJ and ML trees as well as our recombination analysis (SimPlot, RDP) indicate some evidence for recombination in the Sorsum and Karsdorf genomes. This and previous published research, e.g. Patterson et al., 2018, demonstrate that HBV evolution has a complex temporal structure. We therefore used only the network for our main conclusions and do not present a tree topology.

3) The manuscript would also benefit if more detail on sampling were provided. What was the total number of archaeological sites studied, and for each site, the proportion of tooth samples that was human DNA positive and the proportion of these human DNA positive samples that were also HBV DNA positive?

The total number of archaeological sites studied is three (Karsdorf, Sorsum and Petersberg). The total number of screened samples is 45 (three individuals from Karsdorf, 32 from Sorsum and 10 from Petersberg). For Karsdorf and for Sorsum all individuals were positive for ancient human DNA (demonstrating at least 0.1% endogenous human DNA) and showed characteristic substitution patterns. For Petersberg however, only 5 individuals had 0.1% endogenous human DNA and higher. All 5 individuals showed substitution patterns typical of ancient DNA.

4) The authors should put the ancient HBV's discovered into a human history context for the readers, especially since principle component analysis was performed to compare the human DNA genomes recovered in this study with other ancient and modern human genomes (Figure 1—figure supplement 4, the source of these datasets need to be acknowledged). What was the historical background of the people in the region at around the time frames studied? This should be discussed in light of the current state of knowledge about human population genetics and paleogenomics of European people, namely the work of Lazaridis et al., 2014. The authors should also compare the HBV "genotypes" that were present at around 7,000 years ago to those that are present in Germany now.

The principle component analysis of the human DNA was used as additional authentication criteria for the correct dating of the sample. The source of the dataset (Haak et al., 2015, Mathieson et al., 2015) was acknowledged in Materials and methods and we now included the reference in the figure legend of Figure 1—figure supplement 4. As the population genetics is not directly linked to an infection with a particular HBV genotype we did not discuss the population genetic background of each sample in detail. We added a more general section (Results and Discussion, first paragraph) and added two sentences in the Discussion about the potential origin of the Neolithic HBV (Results and Discussion, fifth paragraph). We discussed the Karsdorf and Sorsum genome and the genotypes that are present today in Central Europe in the Results and Discussion section.

5) It has been shown in numerous studies that avihepadnaviruses are able to integrate into their host genomes, via the germ line, to form endogenous copies. Integration outside of hepatocytes has not been demonstrated for HBV. Nonetheless, in the case of this study we would appreciate a rationale for why the ancient HBV sequences represent exogenous virion DNA rather than endogenous elements.

It is unlikely that the found HBV sequences represent endogenous elements of the human genome as this was not shown for modern HBV in humans. In addition, we performed a split read alignment using BWA MEM against HBV and the human genome, by which different parts of the same read can be aligned to different reference sequences. However, we do not detect any clear position of a potential insertion of HBV parts in the human genome. The uneven coverage (see comment 1 above), the high copy number of the found HBV compared to the human genome, and the detected proteins (for Karsdorf and Petersberg) indicate an active exogenous virion and very unlikely an endogenous element.

6) The HBV mutation rate has been a matter of extensive debate over recent years and studies into ancient HBV sequences dating back a few centuries have failed to detect any clock-like signature. These three much older HBV sequences are an exceptional opportunity to assess the evolution of HBV more accurately over time. The manuscript would benefit greatly from some molecular clock analysis.

Previous published research e.g. Patterson et al., 2018, Littlejohn et al., 2016; Simmonds and Midgley, 2005, are describing the problem with molecular clock analysis in HBV. We performed a TempEst analysis (Rambaut et al., 2016) and found that there is very little temporal signal in the HBV data (Figure 2—figure supplement 9). We added a sentence to the main manuscript (Results and Discussion, fifth paragraph). In addition, we find evidence of recombination as described above. We therefore refrain from further dating analysis. We have added the TempEst analysis to the supplementary material.

7) The authors should discuss what significance these ancient HBV sequences have on the origin of orthohepadnaviruses in humans? What changes result from the inclusion of these three ancient sequences in the phylogenetic tree?

As the number of ancient HBV is very low we carefully discussed the significance in the Results and Discussion section.

8) Considering double infection (super/coinfection) with HBV of two different genotypes is a common phenomenon, with prevalence rates ranging from 4-18% (Ding et al., 2003; Kato et al., 2003; Osiowy and Giles 2003; Chen et al., 2004; Olinger et al., 2006), the authors should comment on their certainty that the consensus sequence recovered from each of the ancient samples do represent one HBV strain and not an artificial chimeric HBV sequence, the result of contigs generated from short reads of next-generation sequencing of two different HBV strains circulating within the same infected individual? The authors have discussed the evidence for extensive recombination, but did not discuss whether this may stem from double infections.

For Karsdorf and Petersberg we did not see any indications (e.g. heterozygous positions) for coinfection. The Sorsum alignment shows less than 2% heterozygous positions genome wide. If at all, the low number of heterozygous positions would only suggest an infection with a closely related strain with the same genotype, since there is at least 8% sequence divergence between genotypes. The heterozygous positions could also be explained by accumulated somatic mutations during chronic HBV infections. The parts of the genome that showed signs of recombination in the Sorsum genome did not show any heterozygous positions. The limitations of ancient DNA research, e.g. short reads, postmortem DNA damage and exogenous contamination with microbial DNA can furthermore increase the number of heterozygous positions.